# Nanofiltration as an Efficient Tertiary Wastewater Treatment: Elimination of Total Bacteria and Antibiotic Resistance Genes from the Discharged Effluent of a Full-Scale Wastewater Treatment Plant

**DOI:** 10.3390/antibiotics11050630

**Published:** 2022-05-06

**Authors:** Micaela Oliveira, Inês Carvalho Leonardo, Ana Filipa Silva, João Goulão Crespo, Mónica Nunes, Maria Teresa Barreto Crespo

**Affiliations:** 1Instituto de Tecnologia Química e Biológica António Xavier, Universidade Nova de Lisboa, Avenida da República, 2780-157 Oeiras, Portugal; micaelaoliveira@ibet.pt (M.O.); ines.leonardo@ibet.pt (I.C.L.); tcrespo@ibet.pt (M.T.B.C.); 2iBET—Instituto de Biologia Experimental e Tecnológica, Apartado 12, 2781-901 Oeiras, Portugal; 3Section of Microbiology, Department of Biology, University of Copenhagen, Universitetsparken 15, DK-2100 Copenhagen, Denmark; anafcsilva@gmail.com; 4LAQV-REQUIMTE, Department of Chemistry, NOVA School of Science and Technology, FCT NOVA, Universidade Nova de Lisboa, 2829-516 Caparica, Portugal; jgc@fct.unl.pt

**Keywords:** antibiotic resistance, carbapenem and (fluoro)quinolone resistance, tertiary wastewater treatments, nanofiltration, wastewater reuse

## Abstract

Wastewater reuse for agricultural irrigation still raises important public health issues regarding its safety, due to the increasing presence of emerging contaminants, such as antibiotic resistant bacteria and genes, in the treated effluents. In this paper, the potential for a commercial Desal 5 DK nanofiltration membrane to be used as a tertiary treatment in the wastewater treatment plants for a more effective elimination of these pollutants from the produced effluents was assessed on laboratory scale, using a stainless steel cross-flow cell. The obtained results showed high concentrations of total bacteria and target carbapenem and (fluoro)quinolone resistance genes (*bla*_KPC_, *bla*_OXA-48_, *bla*_NDM_, *bla*_IMP_, *bla*_VIM_, *qnr*A, *qnr*B and *qnr*S) not only in the discharged, but also in the reused, effluent samples, which suggests that their use may not be entirely safe. Nevertheless, the applied nanofiltration treatment achieved removal rates superior to 98% for the total bacteria and 99.99% for all the target resistance genes present in both DNA and extracellular DNA fractions, with no significant differences for these microbiological parameters between the nanofiltered and the control tap water samples. Although additional studies are still needed to fully optimize the entire process, the use of nanofiltration membranes seems to be a promising solution to substantially increase the quality of the treated wastewater effluents.

## 1. Introduction

Water scarcity has been a worldwide problem and a central issue on the international agenda over the last few decades [1,2]. The agricultural sector alone is responsible for the consumption of about 70% of the available freshwater on Earth and this demand is expected to continue to grow, due to the projected increase in the world population in the coming years [3]. This scenario makes wastewater reuse for agricultural irrigation a valuable and sustainable alternative. However, despite this being a practice already implemented in different water-scarce countries around the world [4], there are still important public health issues regarding its quality and safety, even if it complies with the current legislation on water reuse (Regulation (EU) 2020/741), which fails to account for the presence of contaminants of emerging concern, such as antibiotic (AB) resistant bacteria and genes, in the reclaimed water. In fact, multiple studies already point out the inefficiency of the conventionally applied wastewater treatments in the removal of AB resistant bacteria and genes from the treated effluents (for discharge and reuse) [5,6,7,8,9,10,11]. Moreover, their discharge into the environment can degrade the quality of the water bodies, making their subsequent use as a potable water source and for multiple industrial applications difficult [12,13]. Therefore, the use of inappropriately treated reclaimed water for agricultural irrigation purposes may result in the contamination of soils, crops and groundwater reservoirs with these micropollutants, posing a direct risk for both the farm workers and crop consumers [14,15,16].

To improve the quality and safety of the reclaimed water and prevent the harmful effects that may arise from its reuse, several advanced treatment technologies, such as membrane separation processes, ozonation, H_2_O_2_-derived oxidation, electrochemical oxidation and sulfate radical-advanced oxidation processes, have been developed and tested for the removal of different emerging contaminants [17,18,19,20]. Among them, the use of membrane separation processes, such as ultrafiltration, nanofiltration and reverse osmosis, is currently considered a powerful solution, with nanofiltration being one of the most cost-efficient methods to perform enhanced wastewater treatments, since it represents a good compromise between the required water quality and the energy expenditure to produce it [17,18,20,21].

Nanofiltration membranes present separation properties between those of ultrafiltration and reverse osmosis membranes, with a pore size in the order of 1 nm, which corresponds to a molecular weight cut-off in the range of 100 to 5000 Da [22]. Since nanofiltration is expected to be an effective technique for the removal of multiple emerging micropollutants from wastewaters, it can be used to produce high quality effluents in a more sustainable way than reverse osmosis, due to its higher permeate flux and ability to work at lower pressures, which contributes to a decrease in the energy consumption [22,23,24]. In fact, several recent studies that used nanofiltration membranes as a tertiary wastewater treatment technique show promising results regarding the removal efficiencies of different contaminants of emerging concern, such as pharmaceutically active compounds, endocrine disruptors, personal care products and heavy metals [17,20,24,25,26]. However, only a few have already started to address the threat of AB resistance and, in particular, the resistance towards last-line ABs (the last treatment options for patients infected with bacteria resistant to other available ABs), by focusing on the removal efficiencies of both AB resistant bacteria and genes from the treated wastewater effluents [27,28]. Therefore, the main goals of the present work are as follows: (1) to determine the concentrations of total—live and dead—bacteria and to assess the occurrence of carbapenem and (fluoro)quinolone resistance genes in the intracellular and extracellular fractions of discharged effluent samples collected from a full-scale wastewater treatment plant (WWTP); (2) to perform an additional nanofiltration treatment step on these discharged effluent samples using a Desal 5 DK nanofiltration membrane and to further address the removal efficiencies of the total—live and dead—bacteria and of the target carbapenem and (fluoro)quinolone resistance genes in the nanofiltered water samples; (3) to compare the concentrations of total—live and dead—bacteria and of the target carbapenem and (fluoro)quinolone resistance genes present in the intracellular and extracellular fractions of the nanofiltered water samples with those found in the reused effluent samples (produced by the WWTP) and in tap water samples (acting as a control of water that was collected and treated in order to assure high enough quality for direct human and animal consumption).

## 2. Results

### 2.1. Total—Live and Dead—Bacteria Present in the Different Samples

The concentrations of total—live and dead—bacteria present in the different samples are shown in Figure 1 and Appendix A. In the discharged effluent samples, the concentrations of total bacteria were 1.5 × 10^6^ cells/mL–1.1 × 10^6^ cells/mL of live bacteria and 4.0 × 10^5^ cells/mL of dead bacteria (Figure 1, Appendix A). In the reused effluent samples, the concentrations of total bacteria were 8.0 × 10^5^ cells/mL–6.1 × 10^5^ cells/mL of live bacteria and 1.9 × 10^5^ cells/mL of dead bacteria, which represents a significant (*p* < 0.05) logarithmic reduction of 0.28 and a removal rate of 47.02%, regarding the concentrations of total bacteria observed in the discharged effluent samples (Figure 1, Appendix A). For the discharged effluent samples submitted to the nanofiltration treatment—from now on designated as nanofiltered water samples—the concentrations of total bacteria were 1.9 × 10^4^ cells/mL–1.3 × 10^4^ cells/mL of live bacteria and 6.2 × 10^3^ cells/mL of dead bacteria, which represents a significant (*p* < 0.05) logarithmic reduction of 1.89 and a removal rate of 98.72%, regarding the concentrations of total bacteria observed in the discharged effluent samples (Figure 1, Appendix A). For the tap water samples, the concentrations of total bacteria were 8.8 × 10^3^ cells/mL–6.9 × 10^3^ cells/mL of live bacteria and 1.8 × 10^3^ cells/mL of dead bacteria, which represents no significant differences (*p* > 0.05), regarding the concentrations of total bacteria observed in the nanofiltered water samples (Figure 1, Appendix A).

### 2.2. Carbapenem and (Fluoro)Quinolone Resistance Genes Present in the Different Samples

The concentrations of the carbapenem and (fluoro)quinolone resistance genes present in the DNA fraction of the different samples are shown in Figure 2 and Appendix A. All the target carbapenem and (fluoro)quinolone resistance genes—*bla*_KPC_, *bla*_OXA-48_, *bla*_NDM_, *bla*_IMP_, *bla*_VIM_, *qnr*A, *qnr*B and *qnr*S—were detected in the discharged effluent samples (Figure 2, Appendix A). Among them, the most abundant were the *qnr*S and *bla*_VIM_ genes, with concentrations of 5.9 × 10^5^ and 1.9 × 10^5^ gene copy numbers/mL, respectively (Figure 2, Appendix A). Despite the significant (*p* < 0.05) reduction in their concentrations, five of these genes—*bla*_KPC_, *bla*_OXA-48_, *bla*_VIM_, *qnr*B and *qnr*S —were still detected in the reused effluent samples, with concentrations ranging from 3.2 × 10^1^ to 1.2 × 10^5^ gene copy numbers/mL and removal rates between 42.81% and 99.77%, regarding their concentrations in the discharged effluent samples (Figure 2, Appendix A). On the contrary, none of the target carbapenem and (fluoro)quinolone resistance genes were detected in the nanofiltered water samples, with the removal rates regarding their concentrations in the discharged effluent samples being superior to 99.99% (Figure 2, Appendix A). In the control tap water samples, there was also no detection of any of the eight target carbapenem and (fluoro)quinolone resistance genes under study (Figure 2, Appendix A).

With regard to the extracellular DNA (eDNA) fraction of the different samples, the concentrations of the carbapenem and (fluoro)quinolone resistance genes are shown in Figure 3 and Appendix A. Two of the target carbapenem and (fluoro)quinolone resistance genes—*bla*_VIM_ and *qnr*S—were detected in the discharged effluent samples, with concentrations of 1.3 × 10^3^ and 4.3 × 10^2^ gene copy numbers/mL, respectively (Figure 3, Appendix A). Despite the significant (*p* < 0.05) reduction in its concentration, the *qnr*S gene was still detected in the reused effluent samples, with a concentration of 2.8 × 10^2^ gene copy numbers/mL and a removal rate of 34.55%, regarding its concentration in the discharged effluent samples (Figure 3, Appendix A). Similar to what was observed in the DNA fraction, none of the target carbapenem and (fluoro)quinolone resistance genes were detected in the eDNA fraction of the nanofiltered water samples, with the removal rates regarding their concentrations in the discharged effluent samples being superior to 99.99% (Figure 3, Appendix A). In the control tap water samples, there was also no detection of any of the eight target carbapenem and (fluoro)quinolone resistance genes under study (Figure 3, Appendix A).

## 3. Discussion

Membrane separation processes are currently considered among the most promising and attractive solutions for the challenge of water quality and wastewater reuse [29]. In this study, the effectiveness of the nanofiltration technique in the removal of total bacteria and AB resistance genes from discharged effluent samples, collected in a Portuguese full-scale WWTP, was tested in laboratory conditions using a cross-flow system, equipped with a commercial Desal 5 DK nanofiltration membrane.

Since the majority of the environmental bacteria still fails to grow on culture media [30], a culture-independent flow cytometry viability assay was performed for the quantification of the total bacteria present in the different samples, allowing the quantification not only of the non-cultivable bacteria, but also of the dead bacteria. This is especially relevant since dead bacteria can lyse and release their chromosomal DNA and mobile genetic elements to the environment, which may harbor AB resistant genes that can be later assimilated by other bacteria and bacteriophages via natural transformation. The results show that, despite the significant reduction in the concentrations of total—live and dead—bacteria from the discharged effluent to the reused effluent samples (with a removal rate of 47.02%), this reduction was even greater from the discharged effluent to the nanofiltered water samples, reaching a removal rate over 98% and concentrations of total—live and dead—bacteria similar to those observed in the control tap water samples. However, since the Desal 5 DK nanofiltration membranes have a molecular weight cut-off between 150 and 300 Da, it would be expected that all the bacteria present in the discharged effluent samples should be retained by the membrane during the treatment. In fact, the concentrations of total—live and dead—bacteria observed in the nanofiltered water samples can likely be explained by the manipulation of these samples during some of the steps of both their treatment and analysis, where it was not possible to maintain the sterility conditions (for example, after the nanofiltration treatment, when passing through both the permeate collecting tubes and the channels of the flow cytometer). Therefore, as with the tap water samples, which also circulate along the water distribution pipes in non-sterile conditions, it is not possible to obtain a water completely free of bacteria. Furthermore, it is important to mention that the quantifications of total—live and dead—bacteria observed in the discharged effluent samples should only be considered as an indicative microbiological parameter, since not all bacteria harbor AB resistance genes. Despite this, in 2019, more than half of the *Escherichia coli* isolates and more than a third of the *Klebsiella pneumoniae* isolates reported by European countries to the European Antimicrobial Resistance Surveillance Network (EARS-Net) were resistant to at least one of the AB groups under surveillance and the simultaneous resistance to different AB groups was also frequent [31]. Therefore, once in the WWTPs, these bacteria find the suitable conditions to proliferate and horizontally transfer their AB resistance genes to other bacteria [8,32,33,34], leading to an expected increase in the concentrations of AB resistant bacteria and, consequently, a high percentage of these microorganisms in the discharged effluent samples.

Regarding the presence of the target carbapenem and (fluoro)quinolone resistance genes in the discharged effluent samples, their high concentrations reinforce the increasing resistance that the community has been acquiring to these ABs, which is particularly important and worrying in the case of carbapenems, as they are one of the most important groups of last-line ABs [8]. The results obtained for the DNA fraction (the DNA extracted from the bacterial community cells present in the different samples) show that, despite the significant reduction in the concentrations of the eight target carbapenem and (fluoro)quinolone resistance genes, five of these genes—*bla*_KPC_, *bla*_OXA-48_, *bla*_VIM_, *qnr*B and *qnr*S—were still detected in the reused effluent samples. This emphasizes the inefficiency of the conventional wastewater treatments in the removal of AB resistant bacteria and corresponding resistance genes from the treated effluents, which consequently act as important sources of AB resistance dissemination into the environment and back to the human and animal populations. On the contrary, none of the target carbapenem and (fluoro)quinolone resistance genes were detected in the nanofiltered water samples. Their removal rates were calculated considering the previously determined *TaqMan* multiplex qPCR detection limit of 1 gene copy number per milliliter [8] and were superior to 99.99% in all the cases. This complete removal of the eight target carbapenem and (fluoro)quinolone resistance genes shows that the Desal 5 DK nanofiltration membrane is effective in the elimination of multiple AB resistant bacteria and the corresponding resistance genes, highlighting that the bacteria detected in the nanofiltered water samples by the flow cytometry assay were a result of the manipulation of these samples. These results agree with recent studies that evaluated the removal efficiencies of different AB resistance genes from swine wastewaters by nanofiltration [35,36]. As in the present work, the results were promising and verified that, despite the inefficiency of the biological treatments in the removal of these micropollutants, the subsequent nanofiltration treatments led to reductions in the order of 4.98–9.52 logs when compared to raw sewage [35], or higher than 99.79% [36]. In addition, a pilot scale study on the occurrence of multiple AB resistance genes in a municipal wastewater effluent and their treatment by a nanofiltration unit obtained extremely high rejections of these target contaminants [27]. It is also interesting to notice that, in both the discharged and reused effluent samples, the concentrations of total bacteria—1.5 × 10^6^ and 8.0 × 10^5^ cells/mL, respectively—were in the same order of magnitude as the concentrations of some of the target AB resistance genes under study, namely the *qnr*S gene, which presented concentrations of 5.9 × 10^5^ gene copy numbers/mL in the discharged effluent samples and 1.2 × 10^5^ gene copy numbers/mL in the reused effluent samples. At first glance, these results would suggest that a large percentage of the bacteria present in these samples—about 39% of the bacteria present in the discharged effluent samples and 15% of the bacteria present in the reused effluent samples—would harbor at least one of the target carbapenem and (fluoro)quinolone resistance genes. However, it is important to keep in mind that bacteria can harbor not just one, but several plasmids containing resistance determinants. Therefore, multiple copies of the same resistance gene may be located in the same bacteria, which is in fact commonly observed in bacteria harboring plasmids containing *qnr* genes [37,38]. Thus, if on the one hand it is possible that there were fewer bacteria harboring the target carbapenem and (fluoro)quinolone resistance genes in our samples than initially thought, on the other hand there may be more multiresistant bacteria, harboring different plasmids with multiple AB resistance genes. As for the results obtained for the eDNA fraction (the free/extracellular DNA present in the different samples), two of the target carbapenem and (fluoro)quinolone resistance genes—*bla*_VIM_ and *qnr*S—were detected in the discharged effluent samples and one of them—*qnr*S—was still detected in the reused effluent samples. Similar to what was observed in the DNA fraction, the complete removal of the eight target carbapenem and (fluoro)quinolone resistance genes only occurred during the nanofiltration treatment, since none of them were detected in the nanofiltered water samples. These results are in line with a recent study showing that membranes with a molecular weight cut-off smaller than 5000 Da can retain more than 99.80% of the eDNA, both in plasmid and linear forms, with size exclusion as the main retention mechanism [28].

Overall, the results obtained in this study show that the Desal 5 DK nanofiltration membranes have a great potential to be used as a tertiary treatment step in the WWTPs, due to their high removal efficiencies of total—live and dead—bacteria and AB resistance genes. This would allow the production of reclaimed water with superior quality, which could be used not only more safely in the activities where it is already being used, but also in the areas where it is highly needed, such as agricultural irrigation. Nevertheless, additional studies are still required to test the long-term filtration performance of these membranes and to optimize the process under different conditions and contaminant loads. Furthermore, the retentate treatment is also a crucial topic to be addressed in future studies. It might be recycled back to the feed stream of the WWTP without introducing a noticeable charge load on it or, as an alternative, future approaches might consider treating these nanofiltration concentrates with advanced oxidation processes, despite the increased costs to the overall treatment that this option will lead to.

## 4. Materials and Methods

### 4.1. WWTP Description and Sample Collection

The samples were collected from a Portuguese full-scale WWTP designed to treat domestic wastewater of approximately 756,000 population equivalents (P.E.), employing the biological aerated filters technology. After the biological treatment step, most of the produced effluent is directly discharged into the Tagus River, with a smaller fraction being filtered through a cartridge filter, disinfected with the addition of sodium hypochlorite and then reused for green park irrigation and street washing purposes. Three biological samples of 10 L each were collected from these two sampling points—discharged and reused effluents—in sterile containers in July 2020. After collection, all the samples were transported to the laboratory under refrigerated conditions and immediately processed upon arrival. The main steps of the wastewater treatment process are shown in Appendix A and the general analytical control parameters of the discharged effluent samples are listed in Appendix A.

### 4.2. Treatment of the Discharged Effluent Samples with a Desal 5 DK Nanofiltration Membrane

The nanofiltration experiments performed on the discharged effluent samples were conducted using a Desal 5 DK nanofiltration membrane (GE Water & Process Technologies, Feasterville-Trevose, PA, USA) on a laboratory scale, stainless steel cross-flow Sepa CF II Membrane Cell System (GE Water & Process Technologies, Feasterville-Trevose, PA, USA), with an effective membrane area of 54 cm^2^. The Desal 5 DK nanofiltration membrane was selected due to its highly hydrophilic character and low molecular weight cut-off, in order to assure high-water permeability and the rejection of small analytes and/or biological entities. However, other nanofiltration membranes could also have been tested, namely the NF90 (FilmTec, Edina, MN, USA). A scheme of the cross-flow system used in this study is represented in Figure 4, including the Hydra-Cell positive displacement pump, model G-13 (Warner Engineering, INC., Minneapolis, MN, USA), equipped with a variator SEW, used for the circulation of the feed/retentate stream. The installation also comprised pressure transducers installed at the inlet (feed), outlet (retentate) and permeate lines. The permeate flux was determined by measuring the volume of permeate collected in a defined period of time. The temperature was also measured in order to normalize the permeate flux for a reference temperature of 20 °C.

Before use, the nanofiltration membrane was cleaned with distilled water for the removal of any impurities and distilled water was also filtered at 20 bar until a constant flux was achieved, in order to assure an adequate membrane compaction. All experiments were then performed at constant transmembrane pressure (20 bar) conditions and the removal rates of the total—live and dead—bacteria and of the target carbapenem and (fluoro)quinolone resistant genes were calculated using the following equation:(1)$$\mathrm{Removal}\;\left(\%\right)=\left(1-\frac{Cp}{Cf}\right)\times 100$$
where *Cp* and *Cf* are the concentrations of the total—live and dead—bacteria or of the target carbapenem or (fluoro)quinolone resistance gene in the permeate and feed, respectively. During the time course of filtration, the fouling was negligible and the permeate flux was rather constant, at a rate of 230 L/(m^2^∙h).

### 4.3. Detection and Quantification of the Total—LIve and Dead—Bacteria by Flow Cytometry

The samples obtained from both the discharged and reused effluents were primarily filtered in triplicate through sterile 100 µm pore-size nylon membranes (Merck Millipore, Burlington, NY, USA) to remove the larger particles that could interfere with the flow cytometry analysis, whereas the discharged effluent samples submitted to the nanofiltration treatment were directly processed, along with the tap water samples. All the samples were stained in triplicate using the LIVE/DEAD™ BacLight™ bacterial viability and counting kit (Invitrogen, Waltham, MA, USA), which contains both the permeant green-fluorescent SYTO™ 9 dye and the impermeant red-fluorescent propidium iodide (PI) dye to distinguish between the bacteria with intact and damaged cell walls, respectively. Briefly, 1 mL of each sample was incubated with 1.5 µL of SYTO™ 9 and 1.5 µL of PI for about 15 min at room temperature, protected from light, and gently mixed before the analysis. From the SYTO™ 9 versus PI plots, which correspond to FL1 (green channel) versus FL3 (red channel), the gates used for the enumeration of live and dead bacteria were defined. Multiple pure cultures of Gram-positive and Gram-negative bacteria were used to confirm and adjust the defined gates for live and dead bacteria, using fresh cultures and bacterial cells treated with isopropanol. All the experiments were performed in triplicate on a CyFlow^®^ Space (Sysmex Partec GmbH, Gorlitz, Germany), equipped with a blue laser emitting at 488 nm. Single-color controls were performed for instrument adjustment and the aqueous solution Sheath Fluid (Sysmex Partec GmbH, Gorlitz, Germany), used to assure hydrodynamic focusing, was also analyzed with and without staining to measure the background noise.

### 4.4. Detection and Quantification of the Target Carbapenem and (Fluoro)Quinolone Resistance Genes by TaqMan Multiplex qPCR

First, all the samples were filtered in triplicate through 0.22 µm pore-size polyethersulfone filters (Pall Corporation, New York, NY, USA) and the filtration volumes were determined by assuming the clogging of the filters as a measure of, approximately, the same amount of filtered biomass. Therefore, volumes of 50 mL for the discharged effluent samples and 90 mL for the reused effluent samples were filtered. Since there was no clogging of the filters when filtering the discharged effluent samples submitted to the nanofiltration treatment and the tap water samples, the filtered volumes for these two samples were 2000 mL. After filtration, the filters proceeded for DNA extraction and 15 mL of each filtrate proceeded for precipitation and purification of the eDNA. The DNA of each sample was extracted following the standard protocol from the PowerWater Kit (Qiagen, Hilden, Germany), whereas the eDNA was precipitated with absolute ethanol and 3 M sodium acetate, as previously described by the author of [39], and purified using the DNeasy UltraClean Microbial Kit (Qiagen, Hilden, Germany), according to the manufacturer’s instructions. Both DNA and eDNA concentrations and purities were then measured using a NanoDrop 1000 Spectrophotometer (Thermo Fisher Scientific, Waltham, MA, USA). At the end, the detection and quantification of the most clinically relevant and globally distributed carbapenem and (fluoro)quinolone resistance genes was performed, using three previously developed *TaqMan* multiplex qPCR assays [8]. (1) *TaqMan* multiplex qPCR 1, designed for the quantification of *bla*_KPC_ and *bla*_OXA-48_-type genes; (2) *TaqMan* multiplex qPCR 2, designed for the quantification of *bla*_NDM_, *bla*_IMP_ and *bla*_VIM_-type genes; (3) *TaqMan* multiplex qPCR 3, designed for the quantification of *qnr*A, *qnr*B and *qnr*S-type genes. All the experiments were performed in triplicate on a LightCycler 96 Real-Time PCR System (Roche, Basel, Switzerland).

### 4.5. Statistical Analysis

The mean values of the concentrations of the total—live and dead—bacteria and of the target carbapenem and (fluoro)quinolone resistance genes in the different samples were compared using multiple one-way analysis of variance tests (ANOVA). For each test, the homogeneity of the variances was previously evaluated with a Levene’s test. If the resulting differences were significant, the variances were not considered homogeneous and an ANOVA with the Dunnett T3 post hoc test was performed; if the resulting differences were not significant, the variances were considered homogeneous and an ANOVA with the Tukey post hoc test was performed. These statistical analyses were performed using the SPSS 26 software (IBM, Armonk, NY, USA) and the differences were considered significant at *p* < 0.05.

## 5. Conclusions

The practice of wastewater reuse for agricultural irrigation purposes still raises public health issues regarding its quality and safety, due to the inefficiency of the conventional wastewater treatments in the removal of different emerging contaminants from the treated effluents. Among them, the presence of AB resistant bacteria and the corresponding resistance genes in the effluents for discharge and reuse stands out as an important threat in most countries, as these streams are direct gateways for their dissemination into the environment and back to the human and animal populations. This study assessed the potential of a commercial nanofiltration membrane to be used as a tertiary treatment in the WWTPs for a more effective elimination of bacteria and AB resistance genes from the produced effluents. Altogether, the following obtained results showed extremely high efficiency in the removal of total bacteria and AB resistance genes from the discharged effluent samples: (1) the concentrations of total bacteria observed in the nanofiltered water samples were significantly lower than those present in the reused effluent samples and similar to the ones detected in the control tap water samples; (2) the concentrations of carbapenem and (fluoro)quinolone resistance genes in the nanofiltered water samples were reduced for values under the detection limit of 1 gene copy number per milliliter, which is also significantly lower than the concentrations present in the reused water samples and similar to the ones detected in the control tap water samples. Therefore, despite the need for additional studies that test the long-term filtration performance of these membranes, optimize the process under different conditions and contaminant loads and focus on the retentate treatment, Desal 5 DK nanofiltration membranes seem to have great potential to be used as a tertiary treatment step in the WWTPs, allowing the production of high quality reclaimed water to be more safely used in the activities where it is already being used, but also in areas such as agricultural irrigation.

## Figures and Tables

**Figure 1 antibiotics-11-00630-f001:**
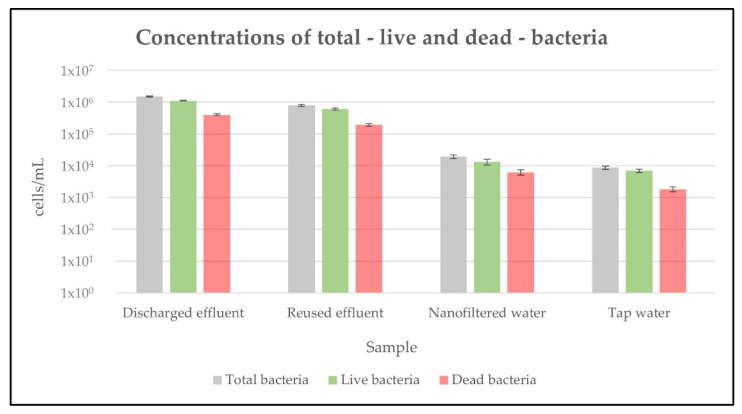
Concentrations of total—live and dead—bacteria in the discharged effluent, reused effluent, nanofiltered water and tap water samples. Values are expressed in cells per milliliter and correspond to the mean ± standard deviation of biological and technical triplicates.

**Figure 2 antibiotics-11-00630-f002:**
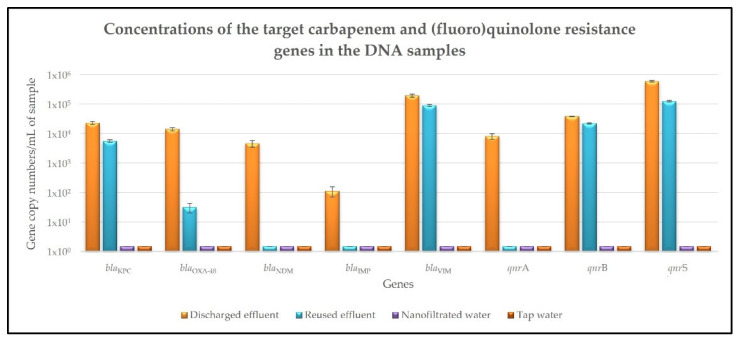
Concentrations of the target carbapenem and (fluoro)quinolone resistance genes in the DNA extracted from the discharged effluent, reused effluent, nanofiltered water and tap water samples. Values are expressed in gene copy numbers per milliliter and correspond to the mean ± standard deviation of biological and technical triplicates.

**Figure 3 antibiotics-11-00630-f003:**
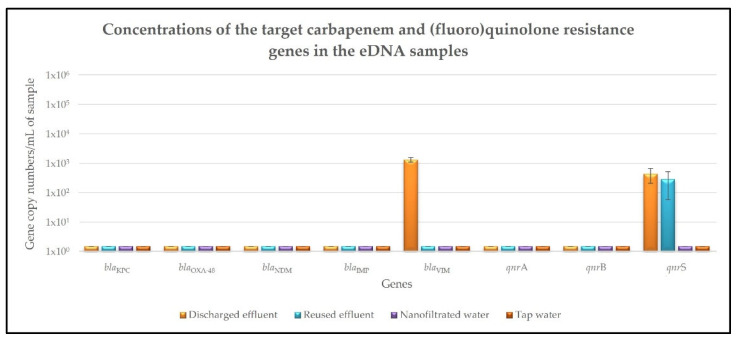
Concentrations of the target carbapenem and (fluoro)quinolone resistance genes in the eDNA precipitated and purified from the discharged effluent, reused effluent, nanofiltered water and tap water samples. Values are expressed in gene copy numbers per milliliter and correspond to the mean ± standard deviation of biological and technical triplicates.

**Figure 4 antibiotics-11-00630-f004:**
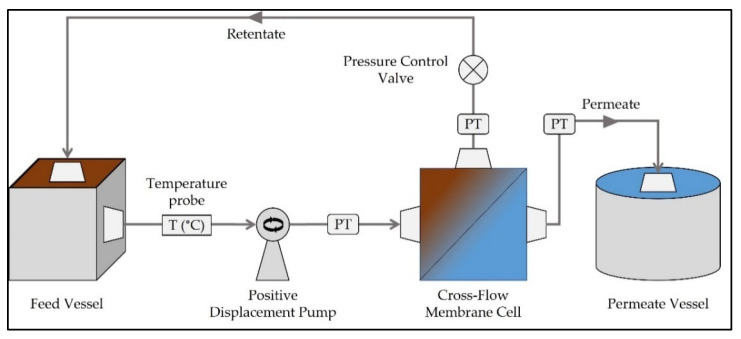
Schematic representation of the cross-flow nanofiltration system used in this study. PT: pressure transducer.

## Data Availability

Not applicable.

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
