# Peer review of "Nanofiltration as an Efficient Tertiary Wastewater Treatment: Elimination of Total Bacteria and Antibiotic Resistance Genes from the Discharged Effluent of a Full-Scale Wastewater Treatment Plant"

_antibiotics, 2022, doi:10.3390/antibiotics11050630_

Round 1
Reviewer 1 Report
This manuscript submitted by Micaela Oliveira et al. evaluate an additional nanofiltration treatment on the discharged effluent samples using a Desal 5 DK nanofiltration membrane to eliminate the total bacteria and antibiotic resistance genes. The results show that the applied nanofiltration membranes might be a promising solution to substantially increase the quality of the treated wastewater effluents. I’d like to recommend this paper for publication if the following issues are addressed.
- The scale of the Y-axis in Figure 3 can be adjusted appropriately to make it clearer.
- Why the authors chose Desal 5 DK nanofiltration membrane? If possible, please compare the differences and advantages of the selected nanofiltration membrane with other commercial nanofiltration membranes.
Author Response
We acknowledge all the suggestions made and hope our responses meet your expectations.
- The scale of the Y-axis in Figure 3 can be adjusted appropriately to make it clearer.
Answer - Thank you for noticing this lapse. The Y-axis of Figure 3 is now adjusted and on the same scale as the Y-axis of Figure 2.
- Why the authors chose Desal 5 DK nanofiltration membrane? If possible, please compare the differences and advantages of the selected nanofiltration membrane with other commercial nanofiltration membranes.
Answer - According to the suggestion, the following information was added to the manuscript: “The Desal 5 DK nanofiltration membrane was selected due to its highly hydrophilic character and low molecular weight cut-off, in order to assure a high water permeability and the rejection of small analytes and/or biological entities. However, other nanofiltration membranes could also have been tested, namely the NF90 (FilmTec, Minnesota, USA)”. - Lines 287-291
Reviewer 2 Report
This manuscript investigated the performance of nanofiltration for elimination of bacteria and ARG from a WWTP, which reported the excellent efficiency of NF. The work is interesting, however, there were some major concerns should be addressed before its publication.
- Although the advantages of NF in bacteria and ARGs were reported, kindly discuss the treatment method of the NF concentrates, as membrane unit is only the separation method without removing the pollutants.
- Line 53, the advanced treatment technologies, such as such as H2O2-derived oxidation, ozonation, sulfate radical-AOP (J Environ Manage, 2022, 304, 114290) should be briefly mentioned.
- For the crossflow membrane cell, how long was operated for obtaining the samples in this study. And how to ensure its representative for long-term operation in the full-scale conditions?
- Kindly suggest discussing the relationship between the number of bacteria and ARGs.
Author Response
We acknowledge all the suggestions made and hope our responses meet your expectations.
- Although the advantages of NF in bacteria and ARGs were reported, kindly discuss the treatment method of the NF concentrates, as membrane unit is only the separation method without removing the pollutants.
Answer - Thank you for the suggestion. The following text was added to the manuscript: “Nevertheless, additional studies are still required to test the long-term filtration performance of these membranes and to optimize the process under different conditions and contaminant loads. Furthermore, the retentate treatment is also a crucial topic to be addressed in future studies: it might be recycled back to the feed stream of the WWTP without introducing a noticeable charge load on it or, in alternative, future approaches might consider treating these nanofiltration concentrates with advanced oxidation processes, despite the increased costs to the overall treatment that this option will lead to.” - Lines 259-266
- Line 53, the advanced treatment technologies, such as H2O2-derived oxidation, ozonation, sulfate radical-AOP (J Environ Manage, 2022, 304, 114290) should be briefly mentioned.
Answer - H2O2-derived oxidation, ozonation, electrochemical oxidation and sulfate radical-advanced oxidation processes are now mentioned in the manuscript, as well as the corresponding reference. - Lines 55-59
- For the crossflow membrane cell, how long was operated for obtaining the samples in this study. And how to ensure its representative for long-term operation in the full-scale conditions?
Answer - The cross-flow membrane cell was operated for 4 hours. During this period, the permeate flux was measured every 30 minutes and the membrane permeance (permeate flux divided by the applied transmembrane pressure) was absolutely constant. During the entire operating time, it was not observed a flux decline. It is true that, in order to assure a permanent operation at constant permeate flux during months, extended pilot tests should be performed. However, these studies are out of the scope of this work, which intended to validate a proof-of-concept.
- Kindly suggest discussing the relationship between the number of bacteria and ARGs.
Answer - Taking into account your suggestion, the following text was added to the manuscript: “It is also interesting to notice that, in both the discharged and reused effluent samples, the concentrations of total bacteria - 1.5 x 106 and 8.0 x 105 cells/mL, respectively - were in the same order of magnitude as the concentrations of some of the target AB resistance genes under study, namely the qnrS gene, which presented concentrations of 5.9 x 105 gene copy numbers/mL in the discharged effluent samples and 1.2 x 105 gene copy numbers/mL in the reused effluent samples. At a first glance, these results would suggest that a large percentage of the bacteria present in these samples - about 39% of the bacteria present in the discharged effluent samples and 15% of the bacteria present in the reused effluent samples - would harbour, at least, one of the target carbapenem and (fluoro)quinolone resistance genes. However, it is important to keep in mind that bacteria can harbour not just one, but several plasmids containing resistance determinants. Therefore, multiple copies of the same resistance gene may be located in the same bacteria, which is in fact commonly observed in bacteria harbouring plasmids containing qnr genes [32,33]. Thus, if on the one hand it is possible that there were fewer bacteria harbouring the target carbapenem and (fluoro)quinolone resistance genes in our samples than initially thought, on the other hand there may be more multiresistant bacteria, harbouring different plasmids with multiple AB resistance genes.” - Lines 227-243
Reviewer 3 Report
The paper is interesting.
I recommend publication only if the following issues can be addressed.
- Equation numbering is missing.
- The authors must discuss the differences between their work and previous studies
- Lines 45-48 page 2: You should mention that discharge of wastewater degrades water quality and thus water cannot be directly used for potable water (via desalination) and industrial applications. I recommend to cite the following references:
Panagopoulos, A., & Haralambous, K.-J. (2020). Environmental impacts of desalination and brine treatment - Challenges and mitigation measures. Marine Pollution Bulletin, 161.
Panagopoulos, A., & Haralambous, K.-J. (2020). Minimal Liquid Discharge (MLD) and Zero Liquid Discharge (ZLD) strategies for wastewater management and resource recovery – Analysis, challenges and prospects. Journal of Environmental Chemical Engineering, 8(5).
- Much more explanations and interpretations must be added for the Results
- How many replications you performed for your experiments?
- Conclusion: Include more of your results.
- Conclusion: Discuss the applicability of your findings/results and future study in this field.
- Language editing is recommended.
Author Response
We acknowledge all the suggestions made and hope our responses meet your expectations.
- Equation numbering is missing.
Answer - Thank you for noticing. The equation is now numbered.
- The authors must discuss the differences between their work and previous studies.
Answer - Comparisons between this study and previous ones were added to the Discussion section. - Lines 219-227; 250-253
- Lines 45-48 page 2: You should mention that discharge of wastewater degrades water quality and thus water cannot be directly used for potable water (via desalination) and industrial applications. I recommend to cite the following references:
Panagopoulos, A., & Haralambous, K.-J. (2020). Environmental impacts of desalination and brine treatment - Challenges and mitigation measures. Marine Pollution Bulletin, 161.
Panagopoulos, A., & Haralambous, K.-J. (2020). Minimal Liquid Discharge (MLD) and Zero Liquid Discharge (ZLD) strategies for wastewater management and resource recovery – Analysis, challenges and prospects. Journal of Environmental Chemical Engineering, 8(5).
Answer - Considering your suggestion, the following sentence and corresponding references were added to the manuscript: “Moreover, their discharge into the environment can degrade the quality of the water bodies, making difficult their subsequent use as a potable water source and for multiple industrial applications [12,13].” - Lines 48-51
4. Much more explanations and interpretations must be added for the Results.
Answer - More explanations and interpretations regarding the obtained results were added over the Discussion section.
- How many replications you performed for your experiments?
Answer - We conducted our experiments with biological triplicates. Then, for each one of them, we performed technical triplicates, making a total of 9 replicates for each experiment, as mentioned in the captions of both figures and supplementary tables.
- Conclusion: Include more of your results.
Answer - More results were included. - Lines 385-393
- Conclusion: Discuss the applicability of your findings/results and future study in this field.
Answer - The Conclusions section was modified according to your suggestion. - Lines 393-399
- Language editing is recommended.
Answer - The manuscript has been revised by an English native speaker.
Reviewer 4 Report
This manuscript (MS) does not have, at the moment, the needed scientific value for being published and this is why I recommend Major Revision based on the following comments:
- The authors should make clear all over the MS the fact that the effluent is generated by a full scale municipal wastewater treatment plant (MWWTP) but it is subjected to a nanofiltration treatment at lab scale or pilot scale ? This is not at all clear, since in the abstract the authors are discussing pilot scale and then in the Discussion chapter they are mentioning laboratory scale.
- The paper objectives are presented but the study novelty, in comparison with international relevant literature is missing.
- The chapter on Materials and Methods should be number 3 (and not 4), because for an experimental study this chapter should precede the Results. Basically, the authors should present in this chapter the following information: the procedures, equipments and reagents to determine the total bacteria and carbapenem and (fluoro)quinolone resistance genes, other procedures to determine the quality indicators of wastewater needed for reuse, the laboratory experimental set-up, operating and washing conditions for nanofiltration.
- There is no description on the general quality of the wastewater effluent resulted from the MWWTP and about the treatment stages. These are information useful especially if the MWWTP has already advanced treatment in place.
- The requirements of the quality standards for the wastewater reuse should be presented in a table containing also the results of this study.
- The first 3 figures have real poor quality and look blurred, they should be replaced.
Round 2
Reviewer 3 Report
Accept in present form.
Reviewer 4 Report
The authors improved their MS based on the comments that I have raised and therefore I consider that this article may be published in its present form.